# Aptamers as Smart Ligands for Targeted Drug Delivery in Cancer Therapy

**DOI:** 10.3390/pharmaceutics14122561

**Published:** 2022-11-22

**Authors:** Zongyi Wei, Yuxin Zhou, Rongjie Wang, Jin Wang, Zhenhua Chen

**Affiliations:** Jiangxi Province Key Laboratory of Drug Design and Evaluation, School of Pharmacy, Jiangxi Science and Technology Normal University, Nanchang 330013, China

**Keywords:** aptamers, smart ligands, targeted, drug delivery, cancer therapy

## Abstract

Undesirable side effects and multidrug tolerance are the main holdbacks to the treatment of cancer in conventional chemotherapy. Fortunately, targeted drug delivery can improve the enrichment of drugs at the target site and reduce toxicity to normal tissues and cells. A targeted drug delivery system is usually composed of a nanocarrier and a targeting component. The targeting component is called a “ligand”. Aptamers have high target affinity and specificity, which are identified as attractive and promising ligands. Therefore, aptamers have potential application in the development of smart targeting systems. For instance, aptamers are able to efficiently recognize tumor markers such as nucleolin, mucin, and epidermal growth factor receptor (EGFR). Besides, aptamers can also identify glycoproteins on the surface of tumor cells. Thus, the aptamer-mediated targeted drug delivery system has received extensive attention in the application of cancer therapy. This article reviews the application of aptamers as smart ligands for targeted drug delivery in cancer therapy. Special interest is focused on aptamers as smart ligands, aptamer-conjugated nanocarriers, aptamer targeting strategy for tumor microenvironment (TME), and aptamers that are specified to crucial cancer biomarkers for targeted drug delivery.

## 1. Introduction

Cancer is one of the top threats to human health. It is estimated that 19 million new cancer cases and 9.9 million cancer deaths occurred worldwide in 2020 [1,2]. Recently, the treatment of cancer is hindered by the complexity and heterogeneity of tumor biology [3,4,5]. In general, most conventional chemotherapeutics will result in nonspecific systemic biodistribution of the drug, inducing cytotoxicity to healthy tissues [6,7]. In addition to poor selectivity, inadequate drug concentration at tumor sites and multidrug resistance generated unsatisfactory therapeutic outcomes. Hence, a targeted drug delivery system is essential to overcome the limitations of current cancer treatment [8,9,10].

Targeted drug delivery systems have advantages over conventional chemotherapy drugs, such as prolonging the circulating half-life of drugs and improving their bioavailability, stability, and tumor accumulation of drugs [11,12,13]. These modifiable physicochemical properties can be used in targeting protocols to ameliorate the biodistribution and target site accumulation of conventional cancer drugs. A targeted drug delivery system usually consists of a nanocarrier and a targeting element [12,14,15]. The targeting element is called a “ligand”. Many ligand molecules, such as bile salts, vitamins, transferrin, saccharides, lectins, antibodies, oligopeptides, and aptamers are currently available for the development of a targeted drug delivery system [16,17,18]. In comparison with conventional small molecule ligands, aptamers have the potential of facile synthesis, simple modification, and exceedingly high specificity in cell-surface aptamer targets, which makes aptamers as an excellent ligand for a targeted drug delivery system [19,20,21].

Aptamers, short RNA or single-stranded DNA, are composed of 20–80 nucleotides, which are considered to be chemical antibodies (Figure 1) [22,23]. Aptamers, acted as pleiotropic ligands, have potential for application in the construction of intelligent targeting systems [24]. The discovery and application of aptamers have been reviewed by previous researchers. There are many reports and studies on aptamer synthesis, aptamer-mediated therapy, and biomedical imaging as a biometric molecule [25,26,27,28]. In recent years, aptamers, acted as targeting agents to modify and bind nanocarriers, have become a hotspot for cancer drug administration and treatment. Some progress has also been made in the research of aptamers that directly bind drugs for tumor targeting, but this review focuses on the application of aptamer-modified nanocarriers as nano-drug delivery systems in cancer therapy. Aptamers are able to efficiently recognize tumor markers such as nucleolin, mucin, and EGFR [28,29,30]. Besides, aptamers can also identify glycoproteins on the surface of tumor cells. Thus, the aptamer-mediated targeted drug delivery system has received extensive attention in the application of cancer therapy [31,32]. This article includes a review of the application of aptamers in targeted cancer therapy, aptamers as smart ligands, aptamer-conjugated nanocarriers, aptamer targeting strategy for tumor microenvironment (TME), and aptamers that are specified to crucial cancer biomarkers for targeted drug delivery are discussed in this work [33,34].

## 2. Aptamers as Smart Ligands

Aptamers are specialized short segments of single-stranded DNA/RNA that have been of increasing interest to researchers in applications since 1990 [35,36,37,38]. The Gold and Szostak labs first described a nucleic acid molecular recognition element. For the first time, the Gold lab described a process that is now known as systematic evolution of ligands by exponential enrichment (SELEX), which identifies one or several molecular recognition elements (MREs) with high affinity and specificity for the intended target [39]. These MREs were later named aptamers [40,41]. The SELEX process usually starts with a chemically-synthesized random oligonucleotide library of different sequences (Figure 2). A single nucleic acid aptamer consists of a specific sequence that evolved from the SELEX program, flanked by two constant primer regions during polymerase chain reaction (PCR) amplification [42,43]. The targets of selection are first incubated under specific ionic and temperature conditions [44]. Molecules that bound to the target are retained and amplified by PCR, while non-bound molecules are discarded.

An aptamer has many advantages. It can bind to specified targets, such as proteins, small molecules, and cells. It also plays a significant role in the recognition of tumor markers, inflammatory factors, and immune cells [45,46]. Compared with other molecular recognizers, the biggest advantage of aptamers is that they have almost no immunogenicity, so they have a very wide range of targeting recognition and binding spectrum. Aptamers are easy to couple to nanocarriers and hardly increase the size of the composite nanocarriers. Aptamers are relatively easy to prepare and store. In comparison with monoclonal antibodies, aptamers have efficient uptake in the presence of specific cell-surface aptamer targets [47,48]. In addition, the establishment of cell-SELEX technology enables aptamers to be used to construct highly targeted cancer nanocarriers (Figure 3) [49]. Cell-SELEX technology was developed in 2003 and uses whole living cells to select aptamers, that are targeted to the cell surface [50]. Cell-SELEX technology and its derivation methods can screen aptamers that can effectively distinguish cancer cells from normal cells. Therefore, aptamers can be used as smart ligands [51,52,53,54].

## 3. Aptamer-Conjugated Nanocarrier Delivery

Nano-targeted drug delivery systems typically consist of a nanocarrier and a targeting component. Nanocarriers have become a promising strategy for tumor-targeted drug delivery due to their unique physical and biological properties. Due to their size characteristics, nano-carriers can enter the blood circulation of the human body. While nanocarriers was loaded drugs (small molecule drugs, protein, genes, and other macromolecular drugs), nanocarriers can slow down drug degradation and renal clearance, increase the half-life of drugs in the blood, and improve their efficacy [55,56]. Due to the increased permeability of tumor neovascularization and its weak lymphatic drainage, nanocarriers can be enriched in tumor tissue to achieve the effect of passive cancer targeting. The selection and preparation of nanocarriers should meet the following requirements: good biocompatibility that includes tissue/blood compatibility and non-immunogenicity; certain mechanical strength as well as be easy to process; and the rate of drug release is moderate and stable. Nanocarriers can be divided into two major categories: inorganic and organic nanocarriers [57,58]. Inorganic nanocarriers mainly include mesoporous silica and magnetic nanoparticles. Organic nanocarriers mainly include polysaccharides, liposomes, and DNA tetrahedron nanostructures (DTNs), which have the characteristics of good biocompatibility and high drug loading rate. Combined with aptamers, nano-delivery systems can improve the specific delivery and precise release of cancer drugs (Figure 4). The drug loading system without aptamer modification only has passive targeting, but non-active targeting. Aptamers that are modified on the surface of nanocarriers can be used as biological recognition molecules for the active targeting of tumor cells. The aptamer-functionalized nanocarriers become intelligent drug carriers with significant drug delivery and targeting properties. The combination of aptamers and nanotechnology has enabled the further application of various targeted drug delivery systems in clinical therapy and diagnosis. Aptamers can bind to nanocarriers in covalent or non-covalent methods. Covalent binding is mainly through the aptamer and the group on the surface of the drug carrier. Non-covalent binding has high affinity and electrostatic interactions [59,60,61]. For instance, Lohiya et al. developed a chitosan-coated (pH-responsive), doxorubicin (DOX)-loaded aptamer -MSNs bioconjugate for the active targeting of breast cancer cells overexpressing epidermal growth factor receptor (EGFR/HER2) [59]. Most aptamer-modified nanocarriers adopt a covalent binding, because they have no additional side effects such as cytotoxicity [62]. The application of nanocarriers combined with aptamers in a targeted delivery system for tumor treatment has attracted extensive attention.

### 3.1. Inorganic Nanocarriers

Inorganic materials have the advantages of nontoxicity, biocompatibility, hydrophilicity, and high stability, and are widely used in the field of drug delivery [63]. The main inorganic nanocarriers include mesoporous silica nanoparticles (MSNs) and magnetic nanoparticles (MNPs), which possess some unique characteristics in drug delivery, including high specific surface area, controllable shape and size, and easy surface modification. Inorganic nanocarriers also have the function of stimulus-responsive drug release [64].

Inorganic nanocarrier-conjugated aptamers are widely used in cancer diagnosis, imaging, and treatment. For instance, DNA aptamer-coupled calcium-phosphosilicate nanoparticles (known as NanoJackets, NJs) are used for the non-invasive detection of prostate and pancreatic tumors [65]. MSNs have attracted much attention in drug delivery due to their high specific surface area, large pore volume, adjustable pore structure, and high surface modifiability [66,67,68]. Similar to other inorganic nanoparticles, MSNs can effectively bind drugs and target drug delivery to the tumor site. As MSNs can protect the encapsulated drug from being recognized by efflux transporters, the drug-carrying MSNs can transport the drug to the cytoplasm after entering the cancer cells, and eventually kill the cancer cells [69]. MSNs can bind the drug EPI and mucin 1 (MUC1) aptamer, demonstrating that MSNs-MUC1-EPI has targeted delivery and a therapeutic effect on MUC1-positive breast cancer cells [70]. The aptamer-MSNs-DOX delivery system is also performed by modifying an aptamer targeting epithelial cell adhesion molecule (EpCAM) to DOX-loaded MSNs. Aptamer-MSN-DOX can be targeted to EpCAM-positive colon cancer cells, improve the efficacy, and reduce toxic side effects [71]. Vivo-Llorca et al. modified MSNs with a MUC1 aptamer to overcome drug resistance for the targeted treatment of triple-negative breast cancer (TNBC) [72]. Aptamer-functionalized MSNs for cancer targeted therapies are listed in Table 1.

MNPs are widely used nanomaterials [81,82], which were first proposed in the 1970s for the clinical targeted treatment of cancer [81,83]. As tumor cells are sensitive to rising temperature, MNPs can release heat through alternating magnetic field stimulation after entering tumor cells, destroying the internal structure of tumor cells to kill cancer cells [84,85,86]. A common strategy is coating poly(lactic-co-glycolic acid) (PLGA) nanoparticles with MNPs. This MNPs system is loaded with a dual drug that is bound by aptamers, which can effectively target the drug to the tumor cells for achieving the targeted therapy effect [87]. Other statements are shown in Table 2.

### 3.2. Organic Nanocarriers

Natural nanocarriers have higher biocompatibility and biodegradability, greater safety, and physiological stability than inorganic nanocarriers. Organic nanocarriers with different stimulation responses have been further applied and developed in the treatment of cancer [64]. Therefore, it has been extensively and deeply studied. Organic nanocarriers mainly include polysaccharides, liposomes, and synthetic polymer nanocarriers [92,93,94]. 

Polysaccharides are natural biopolymers that play different roles in microorganisms, plants, and animals [95,96]. Polysaccharides are polymeric carbohydrates that are composed of at least more than 10 monosaccharides bound by glycosidylic bonds. Polysaccharides that are composed of the same monosaccharides are called isopolysaccharides, such as starch, cellulose, and glycogen. Natural polysaccharides are excellent nanocarriers because of their superior properties with availability, biocompatibility, and extraordinary biodegradability [97]. Due to these excellent properties, natural polysaccharides are widely used in the design of nanocarriers and have a wide range of applications in the delivery and protection of biologically active compounds or drugs [98,99]. Chitosan is a representative kind of matrix. In addition, natural polysaccharides can also be used as a targeted therapy through mechanisms such as specific enzymatic hydrolysis, binding to receptors, pH triggering, and mucosal adhesion [100]. For example, the MUC1 aptamer was used to functionalize chitosan-coated human serum albumin nanoparticles to obtain a selective drug carrier for tumors that were overexpressing MUC1. Besides, chitosan combined with aptamer-coated nanoparticles have strong cytotoxicity and can be used as a potential target tumor drug delivery system [101].

Liposomes are widely used in medicine as nanocarrier systems. Liposomes are vesicles with an aqueous volume that is completely enclosed by a phospholipid membrane. Liposomes range in size from 30 nm to a few microns and can be single or multilayered, where each layer is a bilayer [102]. They play a unique role in combatting drug resistance and improving drug stability [103]. Lipid-based systems include liposomes, solid lipid nanoparticles (SLNs), and cationic liposomes (CLPs). Encapsulation or conjugation of liposome nanocarriers can prolong the half-life of antitumor drugs in blood circulation and significantly improve the stability of drugs in vivo. The aptamer AS1411 was coupled to polyethylene glycolpegylated (PEGylated) CLPs and used as a targeting probe ASLP (AS1411-PEG-CLPs). In addition, the novel siRNA delivery system targeting nucleolar proteins by AS1411 is a potential therapeutic strategy for melanoma [104]. Representative aptamer-functionalized liposomal nanocarriers for cancer targeted therapy are shown in Table 3.

Synthetic polymer nanocarriers mainly include semi-synthetic or synthetic polymers PLGA-PEG, N-(2-hydroxypropyl) methylacrylamide (HPMA), polyacrylamide (PAM), and polyetherimide (PEI), which also have the characteristics of biocompatibility, biodegradation and high drug loading. Polymer nanocarriers with different diameters and shapes can be synthesized or self-assembled by some effective methods for the loading and delivery of different anticancer drugs. There are many reports on the synthesis of polymers by aptamer binding. The binding of polyethyleneimine to gold nanoparticles (AuNPs) that were functionalized with aptamer AS1411 and DOX (PEI-g-PEG) is a promising polymer composite strategy. The prepared AS1411-g-DOX-g-PEI-g-PEG@AuNPs nanoparticles have a diameter of 39.9 nm and can stably exist in water and cellular media, which improves the stability of DOX-AuNPs. In the cell experiment, the cancer cell (A549) can be eliminated [110]. Other studies on aptamer functionalization of synthetic polymer nanocarriers for cancer therapies are listed in Table 4.

### 3.3. DNA Nanostructures as Nanocarriers

DNA nanostructures, which can penetrate cell membranes, are an excellent strategy as nanocarriers. DNA 3D tetrahedron nanostructures have recently generated interest in DNA nanotechnology. DTNs combined with the AS1411 aptamer as capture probes can achieve efficient capture of cancer cells, and this DTNs combined with fluorescent materials can be used for the early diagnosis and clinical treatment of cancer [117]. Gong et al. designed a bimolecular G-tetramer (G4) and adenosine triphosphate (ATP) aptamer as a logical control unit to develop intelligent DNA nano-assembly controlled by YES-AND logical circuits, which has great prospects for intelligent anticancer drug delivery [118]. Specific binding of PD-L1 and Pcsk9 siRNA on well-defined TDNs by DNA hybridization can target the release of immune cells to colon cancer cells and contribute to the treatment of CRC [119]. Other aptamer-targeted cancer therapies that are based on DNA tetrahedral nanocarriers are listed in Table 5.

### 3.4. Disadvantages and Challenges of Nanocarriers

Despite the unique advantages of nanocarriers in the delivery of active anticancer drugs, only a few studies have emerged. For instance, liposomes can improve drug stability in vivo, but they are not stable enough in individual storage, their encapsulation rate is low, and their drug load is small. PEI nanocarriers have made limited progress in clinical practice due to their toxicity and in vivo instability. In inorganic nanocarriers, how to improve the biodegradability of MSNs is an urgent problem to be solved. The preparation process of MNPs is complicated and the particle morphology and structure are not easy to control. In addition, they are vulnerable to oxidation and acid and alkali corrosion during storage. In summary, how to simplify the preparation of nanocarriers, reduce the difficulty of storage, and improve the drug loading rate and biodegradability is still a topic of research in the field of nano-drug delivery systems.

## 4. Aptamer Targeting Strategy for Tumor Microenvironment

Compared with normal tissues, the internal tissues of tumors have a very complex and highly heterogeneous microenvironment due to abnormal metabolism and proliferation at the tumor site [131,132]. Its internal characteristics mainly include abnormal expression of adenosine triphosphate (ATP), glutathione (GSH), and reactive oxygen species (ROS).

During the course of cancer development and progression, ATP, a major metabolite adenosine, and possibly other nucleotides are actively secreted, passively released, or generated in the extracellular environment and play key roles as extracellular messengers. In healthy tissues, the extracellular accumulation of nucleotides and nucleosides is almost negligible. In contrast, ATP and adenosine accumulate at high levels at sites of inflammation and tumors. Thus, the overexpression of ATP can be used as a target that is recognized by aptamers for tumor-targeted drug delivery [133]. For instance, using ATP and MUC1 aptamers that were immobilized on the surface of MSNs. The ATP aptamers leave the surface of MSNs and start drug release after cancer cells have mediated endocytosis through MUC1 receptors [134]. In addition, the AS1411-ATP aptamer chimera can be used as a novel approach to selectively deliver DOX to cancer cells, a strategy that has the potential to increase DOX efficacy and reduce toxicity to normal cells [135].

With respect to cancer, GSH has dual effects in its progression. Excessive reduced glutathione promoted tumor progression, and elevated levels correlate with increased metastasis [136,137]. Thus, glutathione levels can be used to detect cancer progression and can also be used as a delivery target for antitumor drug delivery systems [138,139]. Experiments on human breast cancer cells in vitro showed that glutathione binding RNA aptamer is expected to develop into an effective anticancer and chemotherapy drug [140]. DOX was inserted into an ATP aptamer DNA scaffold and then modified to obtain poly(oxyethylene)-imide (POEI)/DOX/ATP aptamer NPs targeting 3CDIT. The dual responsive release of GSH and ATP made DOX enriched in tumor cells, thus ensuring the effectiveness and safety of glioma chemotherapy [141]. The use of an MNP binding aptamer (MNP/SGc8-SP) to detect glutathione content and expression is a promising and efficient targeted tumor delivery system [142].

ROS are a series of molecules that are produced by intracellular oxidative metabolism, including singlet oxygen (primary excited state), superoxide anion (single-electron state), hydroxyl radical (three-electron state), and hydrogen peroxide (two-electron state) [143,144]. In the TME, a low levels of ROS play important roles in signaling, cell proliferation, and revascularization. The gradual elevation of ROS can also promote tumor cell proliferation and metastasis [145,146]. Over-expressed ROS can damage the DNA of cancer cells, leading to apoptosis and tumor necrosis to a certain extent. Tumor cells have fast growth rate, strong reproductive ability and strong metabolic capacity. Besides, their demand for nutrients is higher than that of normal cells. Therefore, ROS are closely related to various stages of tumors, and the expression level of ROS can be used as a target for aptamer nano-drug delivery systems. For example, phorbol-12-myristate-13-acetate (PMA) was released in HL-60 cancer cells to induce the production of ROS, after the aptamer on MSNs specifically recognized and targeted binding to HL-60 cancer cells. ROS effectively induced apoptosis in HL-60 cells [147]. A priming strategy was developed to selectively kill tumor cells by combining singlet oxygen quenching MnO_2_ with a tumor cell-targeting aptamer. Aptamers on the surface of nanoparticles can recognize proteins on the surface of tumor cells and specifically bind to the induced singlet oxygen production "on-off" switch, which produce high concentrations of ROS at the tumor site to kill the target tumor cells [148].

## 5. Aptamers Specific to Important Crucial Cancer Biomarkers for Targeted Drug Delivery

Cancer biomarkers are molecules that indicate the abnormal cancer status and play important roles in many biological processes, including cell proliferation, cell migration and cell–cell interactions [149,150,151,152,153]. Most cancer biomarkers are proteins [154]. Aptamers with high specificity and affinity for certain tumor surface proteins can be selected from the existing aptamers or DNA/RNA libraries as molecular recognizers to enhance the targeting of nano-drug vectors [155,156,157]. These biomarkers can serve as reliable targets for aptamer drug delivery systems because of the high binding capacity and specificity of aptamers. Specifically, numerous aptamers target cancer-specific signature markers such as human cluster of differentiation antigen 133 (CD133), CD44, and EpCAM. This section lists several representative tumor biomarkers and the corresponding research on aptamer targeting strategies, as well as some other novel research applications (Figure 5).

CD133, a glycoprotein with five transmembrane structural domains that was identified from mouse neuroepithelial stem cells and human hematopoietic stem cells, is a widely recognized tumor stem cell marker [158,159,160,161,162,163,164,165,166]. CD133-coordinated aptamers can be used for the detection of some tumors and targeted delivery of chemotherapeutic agents. The aptamer targeting CD133 was hybridized with partial complementary paired RNA (ssRNA) and modified on the surface of quantum dots (QDs) and AuNPs to construct aptamer nanosensors [167]. PLGA-PEG NPs coupled with a paclitaxel-loaded CD133 aptamer (N-Pac-CD133) were designed to eliminate lung cancer stem cells, and the results showed that N-Pac-CD133 had significantly enhanced targeting and efficacy against lung cancer stem cells [168]. Zahiri et al. prepared a dendritic MSNs-based (DMSN-based), pH-responsive nanoparticle that was functionalized with a CD133 aptamer that was released due to pH changes after endocytosis by tumor cells [169]. A propranolol aptamer-loaded CD133 polylactic acid-glycolic acid copolymer nanoparticle (PPNCD133) was designed for the treatment of infantile hemangioma and showed promising effects on hemangioma [170]. A PEGylated acetylated carboxymethyl cellulose conjugate of SN38 (7-ethyl-10-hydroxycamptothecin) was developed to covalently bind to an aptamer targeting CD133. This nanoplatform was used to specifically deliver SN38 to colorectal cancer cells. Ligand-modified PEG-AcCMC-SN38 nanoparticles with a size of less than 200 nm showed enhanced cellular uptake in CD133-positive HT29 cell lines. In conclusion, the prepared Apt-PEGAcCMC-SN38 can be considered as a promising targeted delivery system for SN38 prodrugs [171].

CD44 is a transmembrane molecule with multiple isoforms that overexpresses in many tumors and promotes tumor formation by interacting with the TME [172,173,174,175,176,177]. CD44 has been implicated in malignant processes including cell motility, tumor growth, and angiogenesis. Alshaer et al. successfully conjugated a 2′-F-pyrimidine-containing RNA aptamer (Apt1) targeting CD44 to the surface of PEGylated liposomes using the thiol-maleimide click reaction. Flow cytometry and confocal imaging were used to detect the uptake of Apt1-Lip by cancer cells. The results showed that the Apt1-Lip was prepared in this article and has higher sensitivity, selectivity and potentiality as a specific drug delivery system [178]. Kim et al. conducted an attractive study of drug-loaded liposomes linked with two DNA aptamers that targeted the surface marker transmembrane glycoprotein MUC1 on breast cancer cells and the surface glycoprotein CD44 antigen on breast cancer stem cells (CSCs). Dual-aptamer-conjugated liposomes (referred to as dual-aptamersomes) were prepared to encapsulate DOX and tested for doxorubicin delivery to 3D cultured breast cancer cells and CSCs. The cytotoxicity of dual-Apt-DOX on CSCs and cancer cells was significantly higher than that of liposomes lacking aptamers [179]. Darabi et al. designed SLNs containing DOX that was decorated with anti-EGFR/CD44 double RNA aptamers. The results indicated that SLNs/DOX/Dexa/CD44/EGFR was a promising new enhanced anticancer delivery system that warranted further clinical trials [109].

An epithelial cell adhesion molecule (EpCAM) is an antigen that is expressed in cancer stem cells and epithelial cells [180]. It was first discovered in 1979 on colon cancer cells [181]. EpCAM is expressed frequently and at high levels in various cancers but at low levels in normal cells [182]. These characteristics make it a biomarker and a therapeutic target for cancer cells [183]. For instance, Zhao et al. developed a cationic liposome-based nanoparticle that was loaded with miR-139-5p (miR-139-5p-HSPC/DOTAP/Chol/DSPE-PEG2000-COOH nanoparticles, MNPs) and surface that was decorated with EpCAM Apt (miR-139-5p-EpCAM Apt-HSPC/DOTAP/Chol/DSPE-PEG2000-COOH NPs, MANPs) for the CRC targeted treatment [184]. In addition, an EpCAM RNA aptamer-conjugated PEGylated liposomal DOX (ER-lip) was also designed for targeted cancer therapy, which demonstrated that ER-lip could promote the survival of animal models and reduce the tumor growth rate. This ER-lip can be used as an ideal drug delivery system for the treatment of tumors with high expression of EpCAM [185]. In the study by Khezrian et al., the surface-active hydrophilic side of functional amphiphilic Janus nanoparticles (JNPs) was functionalized with aptamers against EpCAM to deliver DOX to HT29 cells of metastatic colorectal cancer [186].

## 6. Conclusions

Due to the advantages of DNA/RNA aptamers in biological recognition and binding, they are widely used in intelligent targeted drug delivery, and have become a research hotspot of biological targeting materials. The aptamer derivatives that have been designed and synthesized can specifically recognize tumor markers such as tumor surface glycoproteins. Furthermore, aptamer derivatives play a unique role in the detection of cancer stem cells and the targeted delivery of chemotherapy drugs.

However, the limitations of aptamers in clinical application still exist. To date, the stability and security of the systemic administration of aptamers have not been approved by regulatory agencies such as the FDA. The FDA has only registered one aptamer drug for local administration at the site of action. Furthermore, the affinity and selectivity of aptamers are key factors for targeted drug delivery. Nucleic acid aptamers that are bound or immobilized on nanostructures may change their three-dimensional conformation, leading to changes in affinity and selectivity. In addition, cancer surface biomarkers are highly inconsistent and may be differentially expressed between individuals. Therefore, there is no universal law for the affinity and selectivity of aptamers in cancer therapy, which cannot be determined by a single study expanding into a field. With the progress of technology, these problems will be properly solved.

## Figures and Tables

**Figure 1 pharmaceutics-14-02561-f001:**
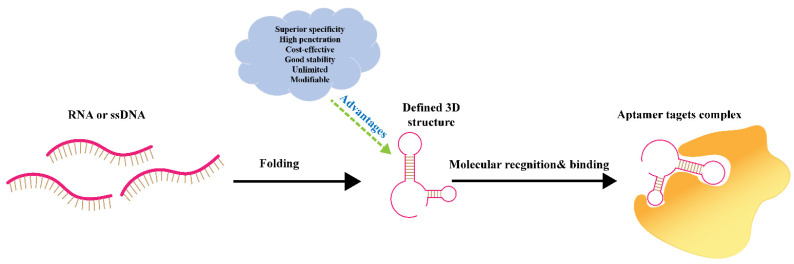
Structure of aptamers and the scheme of molecular recognition by aptamers.

**Figure 2 pharmaceutics-14-02561-f002:**
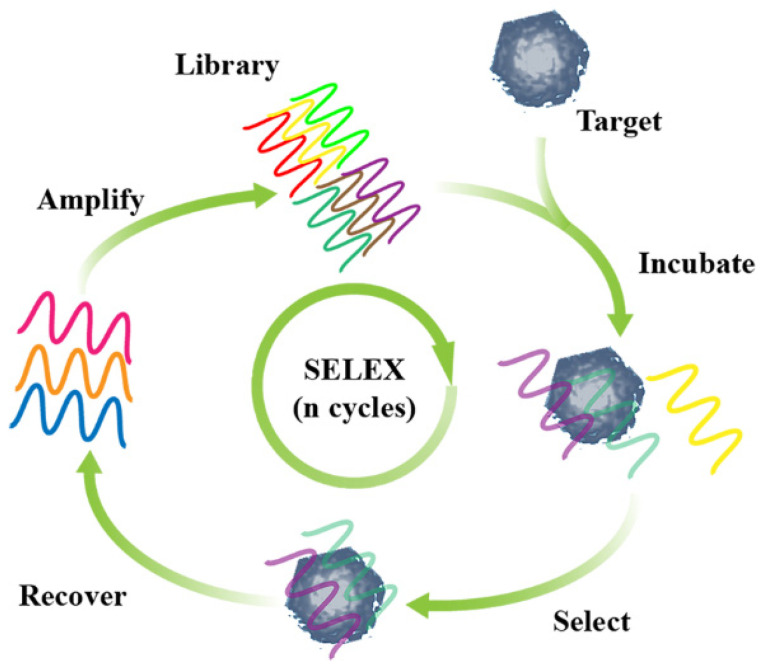
SELEX technology in selection of aptamers.

**Figure 3 pharmaceutics-14-02561-f003:**
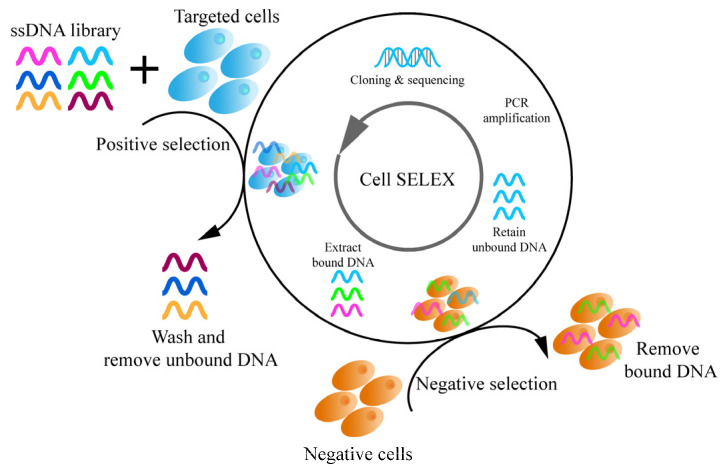
Cell-SELEX technology in selection of aptamers.

**Figure 4 pharmaceutics-14-02561-f004:**
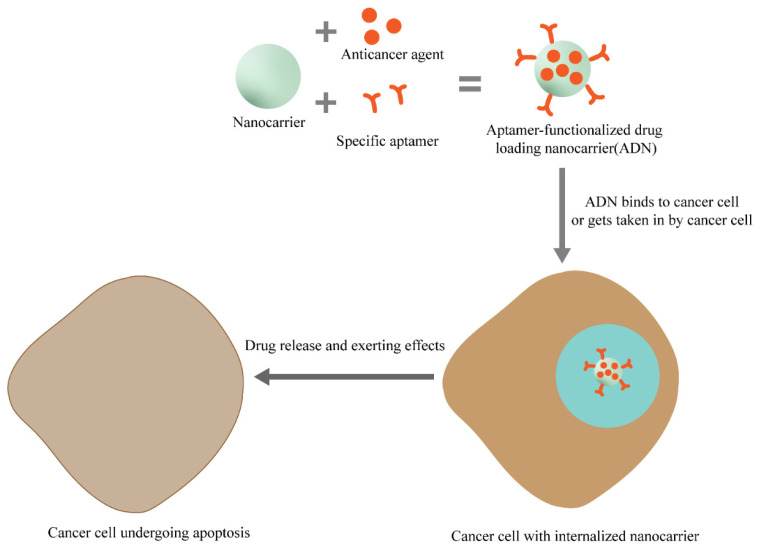
Aptamer-functionalized nanoparticles acting on a cancer cell.

**Figure 5 pharmaceutics-14-02561-f005:**
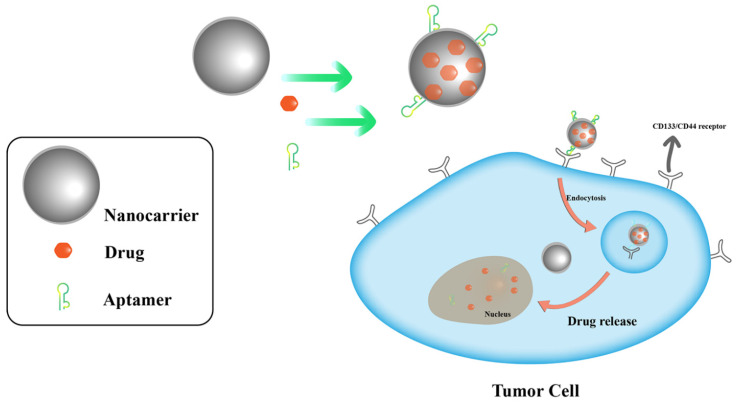
Aptamers recognize biomarker acceptors and release drug.

**Table 1 pharmaceutics-14-02561-t001:** Aptamer-functionalized MSNs for cancer targeted therapies.

Aptamers	Payloads	Cancers	Reference
MUC1	Epirubicin (EPI)	Breast cancer	[70]
EpCAM	DOX	Colon cancer	[71]
MUC1	Navitoclax	TNBC	[72]
MUC-1	Safranin O	Breast cancer	[73]
EpCAM	DM1	Colorectal cancer (CRC)	[74]
HB-5	DOX	Breast cancer	[75]
Sgc-8	DOX	Human acute T-lymphocyte leukemia	[76]
AS1411	DOX	MCF7 cells	[77]
EpCAM	DOX	Human HT-29 tumors	[78]
EpCAM	miR-328	SW480 cells	[79]
NCL-aptamer	DOX	Breast cancer	[80]

**Table 2 pharmaceutics-14-02561-t002:** Aptamer-functionalized MNPs for cancer targeted therapies.

Aptamers	Payloads	Targets	Reference
AS1411	Curcumin and gemcitabine	Pancreatic cancer	[87]
AS14 and AS42	Aptamer-modified FeAG nanoparticles	Ehrlich carcinoma cells	[88]
Sgc-8	DOX	Lung adenocarcinoma cells (A549)	[89]
Mucin16 (MUC16)	Erlotinib	Ovarian cancer cells	[90]
Vascular endothelial growth factor (VEGF) DNA	VEGF DNA aptamer	Ovarian cancer cells	[91]

**Table 3 pharmaceutics-14-02561-t003:** Aptamer-functionalized liposome nanocarriers for tumor targeted therapies.

Aptamers	Nanocarriers	Payloads	Cancers	References
AS1411	PEGylated CLPs	Anti-BRAF siRNA (siBraf)	Melanomas	[104]
AS1411	PEGylated SLNs	Docetaxel (DTX)	CRC	[105]
A15	PEGylated SLNs	Oxaliplatin (OXA)	Hepatocellular carcinoma (HCC)	[106]
AS1411	PEGylated liposome	Paclitaxel (PTX)	Renal carcinoma	[107]
A15	CLPs	PTX and Survivin siRNA	Brain glioma	[108]
anti-CD44 and EGFR aptamers	SLNs	DOX	Breast cancer	[109]

**Table 4 pharmaceutics-14-02561-t004:** Aptamer-functionalized synthesis polymer nanocarriers for cancer therapies.

Aptamers	Nanocarriers	Payloads	Cancers	References
MUC1	PLA-PEG	DOX	Lung cancer	[111]
EpCAM	Poly(amino acid)s NPs	Tanshinone II-A (TSIIA)	CRC	[112]
AS1411	PA-Apt-CHO-PEG	PA	Breast cancer	[113]
CD30	PEG-PLGA	DOX	anaplastic large cell lymphoma (ALCL)	[114]
A10	PLGA	Triplex forming oligonucleotides (TFO)	Prostate cancer	[115]
AS1411	PEI	5-fluorouracil (5-FU)	Gastric cancer	[116]

**Table 5 pharmaceutics-14-02561-t005:** Aptamer-functionalized DNA tetrahedron nanocarriers for cancer therapies.

Aptamer	Payloads	Cancers	References
Aptamer cluster	DOX	Breast cancer	[120]
AS1411	DOX	HeLa and 4T1 cells	[121]
AS1411	Cy3 and Cy5	Breast cancer	[122]
anti-HER2	Maytansine	HER2-positive cancer	[123]
anti-MUC1	DOX	Breast cancer	[124]
Gint4.T	DOX	Glioma	[125]
Pegaptanib	Pegaptanib	Oral cancer cell	[126]
AS1411	5-FU	Bbreast cancer	[127]
MUC1 and AS1411	DOX	Breast cancer	[128]
SL2B	DOX	CRC	[129]
MUC1	DOX	Breast cancer	[130]

## Data Availability

Not applicable.

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
