# Peer review of "Aptamers as Smart Ligands for Targeted Drug Delivery in Cancer Therapy"

_pharmaceutics, 2022, doi:10.3390/pharmaceutics14122561_

Round 1
Reviewer 1 Report
The authors review the potential role of aptamers in drug targeting, across different systems. The review is interesting, well-written and up-to-date. However, there are some minor errors that should be corrected before publication. I indicate some of them below:
Page 5. the phrase "most of cancer biomarkers are proteins" should begin with a capital letter.
Page 7. In my opinion, the sentence "TNF-α, a member of the tumor necrosis factor superfamily, is a marker on the surface of a variety of tumors" is not correct. TNF alpha is a soluble factor.
Reviewer 2 Report
The manuscript by Zongyi Wei et al reviews aptamers as smart ligands for targeted drug delivery in cancer therapy. The paper describes existing strategies of aptamer assisted drug delivery to cancer cells. The research field authors selected for review is indeed quickly developing and is of immediate interest. However, the paper contains a large number of shortcomings detailed below.
Authors claim the lack of reviews in selected field. In fact, brief scanning of pubmed revealed a large number of recent review papers that considerably overlap with the manuscript. The most recent good example may be found here: https://doi.org/10.1016/j.bioadv.2022.213077
Other examples of review papers that are close by content: 10.1016/j.addr.2018.08.005; 10.3390/ijms21239123; 10.3389/fbioe.2022.972933; 10.2174/0929867325666181008142831; 10.1080/1061186X.2018.1491978.
Authors consider preferably targeted drug delivery systems that include a nanocarrier, like MSNs, MNPs etc. However, nanocarrier-free aptamer-based delivery systems have been reported as well (for example, see ref 120).
Figure 1 gives only two examples of many possible secondary structures. The inverted part of steem-loop structure looks strange. Designation of nucleobases in figure legend does not seem appropriate.
DNA nanostructures are known to penetrate the cell membrane and have been used as nano-carriers. Few papers described aptamer-based scaffolds of this type. Particularly, aptamer/DOX/DNA tetrahedron scaffold conjugates have been used to target cancer cells. This aspect has not been considered in the review.
The legend to fig 2 refers to (a) SELEX and (b) cell-SELEX. Only one panel is shown in the figure.
The coverage of the sources is poor. As an example, the search of pubmed for “mesoporous silica nanoparticles cancer aptamer” resulted in 65 papers. 17 matching papers (only part of all matching papers) were selected. Of these 17 papers (listed below) only 3 were found in the cited literature:
MSNs:
10.1080/03639045.2017.1371734; 10.1002/chem.202001579; 10.1016/j.nano.2022.102588; 10.2147/IJN.S295740; 10.1016/j.ejps.2015.12.014; 10.3109/1061186X.2012.761222; 10.1016/j.nano.2017.08.006; 10.2147/IJN.S143293; 10.1016/j.actbio.2015.01.002; 10.1002/adhm.201200116; 10.1186/s11671-019-3208-3; 10.1080/10837450.2019.1569678; 10.2147/IJN.S278724; 10.1186/s12951-021-01056-3; 10.1021/ja110094g; 10.1039/d0tb01960g; 10.2217/nnm-2017-0353.
TRAIL on page 7 needs brief description on how mir-212 promotes apoptosis.
The text has to be checked for proper formatting: capital letters in the middle of sentences and small letters at the beginning in few cases. English certainly needs proofreading.
Reviewer 3 Report
The manuscript review entitled "Aptamers as smart ligands for targeted drug delivery in cancer therapy" from Wei et al involved the bibliographical analysis of aptamers as ligands for targeted drug delivery systems in cancer therapy.
The introduction is according to the developed topic of the manuscript. Unfortunately, there are current published material that widely covers this interesting topic.
Some of the missing references related to this important point are:
Theranostics 2015; 5(4):322-344. doi:10.7150/thno.10257. Aptamers: Active Targeting Ligands for Cancer Diagnosis and Therapy
ACS Nano 2010, 4, 3, 1433–1442. https://doi.org/10.1021/nn901374b. Aptamer-Based Tumor-Targeted Drug Delivery for Photodynamic Therapy
J. Am. Chem. Soc. 2022, 144, 4, 1493–1497. https://doi.org/10.1021/jacs.1c09574. Aptamers Entirely Built from Therapeutic Nucleoside Analogues for Targeted Cancer Therapy
J Control Release 2013 Oct 28;171(2):152-62. doi: 10.1016/j.jconrel.2013.06.006. Smart ligand: aptamer-mediated targeted delivery of chemotherapeutic drugs and siRNA for cancer therapy
TrAC Trends in Analytical Chemistry. Volume 82, September 2016, Pages 316-327. https://doi.org/10.1016/j.trac.2016.06.018. Aptamers as smart ligands for nano-carriers targeting
Additionally, it should be necessary to find a different and complementary point of view of this topic to improve the information regarding the previous published reviews.
Furthermore, the authors could amplify the information in each sub-topic including, as an example in the inorganic nanocarriers, one figure or table with all the information, biological activity figure of these nanocarriers, etc. This suggestion is important for each topic the authors developed.
Also, in order to improve the manuscript and as a suggestion to increase the interest of the potential readers, it is significant to clarify advantages and disadvantages of each selected aptamer system.
Finally, I invite the authors to include the abbreviation list of words at the end of the article.
Round 2
Reviewer 2 Report
The paper was substantially improved by the authors; however few points of concern are still present.
First, I would like to repeat the point 2 of my previous review, as it was not taken correctly by the authors. “Nanocarrier-free aptamer-based delivery systems have been reported in the literature”, but have not been reviewed in the paper. This means the use of aptamer-drug conjugates WITHOUT nanocarrier. It does not mean aptamer-free nanocarrier. If the review of aptamer-drug conjugates without nanocarrier was not originally planned, I would suggest adding a sentence somewhere in the Introduction that the paper primarily focuses on nanocarrier-based methods.
Additional points relate to missing references to some statements, poor English expressions, and incorrect statements. More details are given below.
Incorrect statements
1. Page 1. “excellent tissue permeability”. Page 3. “efficient tissue uptake”. I suppose, authors refer to the aptamers that target cell-surface exposed targets. Generally, oligonucleotides do not show efficient uptake. I would suggest adding a phrase that refers efficient uptake to the presence of specific cell-surface aptamer targets.
2. Page 2. “SELEX process usually starts with an enriched random library of different molecules (Figure 2).” SELEX usually starts with an unenriched random library. Enrichment is a result of selection rounds.
3. Page 2. “single nucleic acid aptamer consists of two constant regions attached to primers during PCR amplification”. Single aptamer consists of specific sequence evolved from SELEX procedure and flanked by the two constant primer regions (needed for PCR amplification).
4. Page 6. “Composed of similar types of long monomers or combinations of other monomer chains.” What is “long monomer”? Chain composed of monomers is a polymer.
5. Fig. 1. “Structure of aptamers”. More appropriate legend would be something like “the scheme of molecular recognition by aptamers”. Please, clarify what “Unlimited” means in advantage cloud.
6. Fig. 3. Does cloning and sequencing precedes PCR amplification?
7. Table 1 in row 2 “DNA” should read EpCAM.
Missing references
1. Page 4. “..nano-carriers can slow down drug degradation and renal clearance”. Ref. to slow down.
2. Page 4.”.. Aptamers binding to nanocarriers in covalent or non-covalent methods.” Ref. to non-covalent as an example.
3. Page 4. “Most aptamer-modified nanocarriers adopt covalent binding mode declining additional side effects, such as cytotoxicity” Ref. to decline of cytotoxicity by covalent binding.
4. Page 5. “Inorganic nanocarriers also have the function of stimulus-responsive drug release” Ref. to stimulus-responsive drug release.
5. Page 7. “..which improves the stability of DOX-AuNPs” “In the cell experiment, the cancer cell (A549) can be eliminated” Ref. to this paper(s).
6. Page 9. “The aptamer targeting CD133 was hybridized with ssRNA and modified on the surface of QDs and AuNPs, respectively, to construct aptamer nanosensors.” Is it different from [159]? Please, ref. to the paper.
Poor English expressions and formatting
1. Page 1, in abstract “ Therefore, Aptamers have potential..” Capital in “Aptamer “
2. Page 1, in abstract and Page 2 “Besides, aptamers can also identify glycoproteins and cell membrane pathways on the surface of tumor cells.” Aptamers can identify pathways on the surface?
3. Page 1, “These modifiable physicochemical properties can be used in targeting protocols to ameliorate the biodistribution and target site accumulation of free drugs, which was contributed to improving the specificity of conventional cancer.”
4. Page 3, “Aptamers have many attractive advantages in high degree specificity and efficient tissue uptake, which in comparison of monoclonal antibodies”
5. Page 3, “…from normal cells can be screened by Cell-SELEX technology and..” Capital in “Cell”
6. Page 4, “non-immune antigenicity”, correct?
7. Page 4, “After combined with aptamer, the surface modified aptamer of nanocarriers, which can be used as a biometric molecule to actively target tumor tissue.”
8. Page 4, “…and nanotechnology has got to further research…” has led?
9. Page 4, “Aptamers binding to nanocarriers in covalent or non-covalent methods.” Bind?
10. Page 5, “Inorganic nanocarriers main include…” main inorganic nanocarriers include?
11. Page 6, “Composed of similar types of long monomers or combinations of other monomer chains.” They composed of similar types of monomers or their combinations in polymer chains?
12. Page 6, “Liposomes as nanocarrier systems are universal used in the medical field”
13. Page 7, “There are many reports on the synthesis of Polymer by aptamer binding.” Capital in Polymer.
14. Page 8, “Despite the unique advantages of nanocarriers in the delivery of active anticancer drugs, some mature studies have emerged” May be, have not emerged? Or only few emerged?
15. Page 8, “Vulnerable to oxidation and acid and alkali corrosion during storage.” They?
16. Page 9, “sinle-let” singlet
17. Page 10, “A2'-f-pyrimidine-containing RNA previously screened for an aptamer against CD44 Apt1 was successfully used to bind the anti-CD44 aptamer to Apt1-Liposomes and tumor cells expressing CD44 using sulfhydryl-linked to the surface of polyethylene glycolized liposomes” Not clear.
18. Page 10, “EpCAM is expressed frequently and at high levels in various cancer detection” Detection?
19. Page 10, “The surfactant hydrophilic side of the Janus platform and the aptamer of EpCAM, which…” predicate is missing.
20. Page 11, “Therefore, there is no universal law for the affinity and selectivity of aptamers in cancer therapy that cannot be determined by a single study expand into a field.”
Reviewer 3 Report
The authors performed all the suggested corrections and additions.
Author Response
Thank you for your suggestion and advises.